# Moving to productivity: The benefits of healthy buildings

**Juan Palacios[1,2], Piet Eichholtz[3], Nils Kok [3]***

**1** Center for Real Estate, Department Urban Studies and Planning, Massachusetts Institute of Technology, Cambridge, Massachusetts, United States of America, **2** IZA, Bonn, Germany, **3** Department of Finance, School of Business and Economics, Maastricht University, Maastricht, The Netherlands

* n.kok@maastrichtuniversity.nl

**Data Availability Statement:** The data are available from the Harvard Dataverse at https://doi.org/10.7910/DVN/ALUUEC.

**Funding:** The Nederlandse Organisatie voor Wetenschappelijk Onderzoek provided funding to Nils Kok for this research – VIDI tel:452-13-004. The research project also received financial support

## Abstract

Health is a critical factor for the generation of value by workers. Companies bear substantial costs associated with absenteeism and presenteeism among their employees. This study investigates the impact of the environmental conditions in the workplace on the health and job satisfaction of employees, as core factors of productivity. We provide evidence based on a natural experiment, in which 70% of the workforce of a municipality in the Netherlands was relocated to a building with a design focused on sustainability and health and well-being. We construct a longitudinal dataset based on individual surveys of the entire municipality work-force and include measures before and after the move. The estimation results show a significant improvement in the perceived environmental conditions, as well as in the health and well-being of the relocated workers, measured by the drop in incidence of sick building syndrome symptoms. Results are heterogeneous based on age: older groups of employees enjoy larger health impacts. The relocation effects remain persistent in the medium term (two years after the moving date). Importantly, a mediation analysis suggests that the achieved improvements in health and well-being lead to significantly enhanced job satisfaction and a 2% reduction in the prevalence of sick leave.

## 1 Introduction

Workers represent a critical input factor for the modern firm, but our understanding of the effects of workplace environmental conditions on human performance is limited. Companies bear substantial costs in the form of both absenteeism and presenteeism, that is, productivity losses due to workers not being able to work at full capacity [1]. Based on a sample of 28,902 working adults in the U.S. [2] document that 13% of the total workforce experienced a loss in productive time due to common pain conditions such as headaches or back problems. The authors estimate a loss of $61.2 billion per year due to pain-related lost productivity.

The literature provides some evidence on the harmful effects of indoor environmental quality (IEQ) in the workplace on employee productivity. Poor indoor air quality in the form of high levels of $CO_2$ or pollutants has been linked to the prevalence of absenteeism, sick building syndrome (SBS) symptoms [3], and reduced cognitive performance of workers [4; 5].

from the INTERREG Healthy Building Project. The other authors received no individual, specific funding for this work.

**Competing interests:** The authors have declared that no competing interests exist.

Inadequate thermal conditions in the form of suboptimal temperatures or relative humidity have been linked to the prevalence of increased heart rates, respiratory problems, SBS, and reduced cognitive performance [6; 7]. Noise is also a risk factor commonly found in workplaces. The exposure to unhealthy decibel levels leads to cardiovascular disease, stress, and sleep disruption, ultimately harming employees' cognitive performance and labor productivity [8]. Finally, light quality has been linked to eye-irritation problems and changes in the circadian rhythm of adults [9].

Most of our understanding about the impact of indoor environmental quality on employees is based on occupant surveys. Recent reviews of the literature from [10] and [11] provide a comprehensive overview of the literature investigating the benefits for occupants of being located in so-called "green", environmentally certified buildings. In both reviews, the results from the majority of the surveyed studies indicate that individuals in "green" buildings evaluate their perceived health and the environmental conditions at their workplace as "better" or "improved." However, these studies are based on cross-sectional comparisons of the answers of participants working in "green" buildings with the answers from those working in conventional buildings. The validity of the results therefore relies on strong assumptions about the differences between employers and employees in "green" and "non-green" buildings. The studies rely on assuming an absence of selection bias, which would arise if the health and working conditions of occupants in sustainable buildings would differ from the health and working conditions of those working in conventional buildings, beyond the building infrastructure.

[12] is currently the only study in the field that is based on a longitudinal design. The authors followed a group of 262 individuals who moved from a conventional to a sustainable building in Michigan. The authors found that study participants reported a significant improvement in the perceived environmental conditions at their workplace, after the move. This resulted in a reduction in the number of hours absent from work due to health reasons, and a higher perception of individual productivity. However, the study has some limitations. First, the post and pre-surveys were implemented in different seasons, when the presence of allergies, and other diseases might differ, as well as the impact of outdoor climate conditions on the indoor environment. In addition, the study lacks a control group that serves as a benchmark for changes in the general health status of employees in the organization over time. Finally, some of the participants in the study were asked to fill out the questionnaire retrospectively, after the move took place.

This paper is based on the variation created by a natural experiment, in which 70% of the workforce of a large municipality in the Netherlands was relocated from a set of conventional buildings to a building designed with sustainability and health and well-being principles in mind. The remaining 30% stayed in their original workplaces over the entire study period. We develop a unique dataset monitoring the perceived working conditions, health, and job satisfaction of more than 600 municipality workers up to two years after the relocation. In total, we surveyed the employees four times, once before the move and three times after the move. We employ a traditional difference-in-differences (DiD) approach to estimate the impact of the move on perceived working conditions and employee health. We evaluate changes in (1) employee-perceived environmental conditions in their workplace, and (2) health outcomes, measured by the presence of SBS symptoms. The results indicate that the relocation led to significant improvements in working conditions, especially in air quality, and health outcomes, with reduced SBS symptoms.

The literature documents significant discrepancies between the short- and long-term reported impacts of individuals associated with material upgrades in their lives due to hedonic adaptation, a psychological process that attenuates the long-term impact in conditions. For

instance, individuals even adapt to serious chronic health conditions (i.e., disabilities), exhibiting high levels of happiness or life satisfaction close to the baseline level, in the long term [13]. A recent study shows this adaptation also appears to exist when evaluating the impact of major building infrastructure improvements. Two years after a major housing retrofit, [14] found occupant well-being levels were identical to the baseline levels, i.e. before the intervention. In the second part of the study, we therefore decompose the estimates of the three surveys administered in the two years after the moving date, to investigate the differences between short and long-term effects. The estimates of health and perceived environmental quality show a strong persistence over time.

Finally, we estimate the impact of the relocation on two measures of productivity: employee satisfaction and the extent to which employees take sick leave. To measure this effect, we estimate a meditation analysis [15], in which the presence of SBS symptoms is the mediating variable between perceived environmental conditions on the one hand, and job satisfaction and actual sick leave on the other hand. The results show that the move to the green, healthy building significantly enhanced job satisfaction, and reduced the prevalence of sick leave by 2%. With an annual salary spent of €54 million at the new municipal building alone, this relative small effect has substantial benefits for the public finances of the municipality. Importantly, these "co benefits" are additive to the direct cost savings on utilities and maintenance, which, according to the calculations of the municipality, already outweigh the marginal construction costs by far.

The results reported in this paper have implications for organizations in their considerations of leasing or buying (office) space. Typically, those considerations do not include the health aspects of a building (e.g. ventilation rates, access to natural light, material use), but our research shows that such factors affect perceived well-being, and ultimately two proxies for productivity. For developers and owners of real estate, energy efficiency is increasingly on the radar, but our research shows that other aspects of sustainability, including the extent to which a building incorporates aspects of health and well-being, are important for users. As the salience of such aspects increases, it is likely that rents and occupancy rates will differentiate based on health and well-being, similar to what has been observed for energy efficiency and sustainability labels [16; 17]. Finally, our results suggest that policy makers should more actively consider buildings as part of policies aimed at prevention of (chronic) disease. Not only do individuals spend 90% of their time indoors—these same buildings have a significant impact on measurable health outcomes, and the demand for health care. Designing regulation or other public policies to measure and improve the health aspects of buildings may be an efficient means to reducing the burden of health care on society, now and in the future.

The remainder of our paper is organized as follows. In the next section, we discuss the literature investigating the link between environmental conditions in offices and employee health. In section 2, we provide a description of the relocation event and the research design used in the study, and we also explain the construction of our health and environmental-quality survey-based indicators. In section 3, we present our empirical strategy. The results are presented in section 4, and section 5 concludes.

## 2 Study set-up

### 2.1 Background

In 2016, Venlo, a municipality in the southeast of the Netherlands, inaugurated a newly constructed office building for use by the municipality. The new municipal building was built following "green" and sustainable principles: In addition to glass and concrete, the north wall of the building is covered with vegetation, and includes a green wall of 2,000 $m^2$. Green or living

walls allow plants to grow from the vertical structure. The installation of green walls has been associated with an improvement in outdoor and indoor air quality, transforming carbon dioxide ($CO_2$) into oxygen, and filtering fine particles from outdoor sources of pollution [18]. In addition, the plants serve as natural insulation against heat, cold, and sound [19]. The building is also equipped with state-of-the-art natural ventilation technology. The air enters the building at the top, where it is oxygenated by plants and brought to the bottom of the building, from where the purified air then circulates naturally throughout the building using physical principles rather than mechanical ventilation systems.

While the new building was designed to the highest standards of green building principles, it has not applied for a green building certificate such as BREEAM or LEED. In the very first stages of the building's design process, the aim was to opt for BREEAM certification. However, at that time, BREEAM did not incorporate building health aspects in its certification process and was solely focused on energy efficiency, and, for example, recycling of building materials. This would have steered the design into an unwanted direction: not enough ventilation (to improve energy efficiency) and recycled, but potentially unhealthy building materials. On top of that, the certification process was deemed expensive, and the municipality thought that money would be better spent on a high-quality interior climate maintenance system, which was indeed installed. The new building has been certified under the "Cradle to Cradle" principle, an attestation to the re-usability of products in the building. In addition, it has won a range of accolades for its sustainable design and performance [20].

In the summer of 2016, 70% of the 1,461 workers of the municipality were moved to the newly constructed office building, within the same city. A round of interviews with high-level officials from the municipality confirms that the selection of "movers" was quasi-random, through selection of teams (or: functional groups), rather than individuals. In other words, relocation decisions were independent from the performance, health, or well-being of individual employees. Three well-defined functional groups did not move. First, all city employees who worked in social services and welfare provision were already located in a separate building before 2016 and stayed there. Second, the complete department of public works, dealing with parks, playgrounds, green maintenance, roads, traffic lights, etc. was housed at a separate location before 2016 and stayed there. Third, in 2015, one year before the new City Hall became operational, the city of Venlo completed the renovation of an adjacent building, previously an industrial site. This building does not have a public entrance or reception desk, which is why the municipality decided to locate the Finance and IT departments in the building, since these departments are mostly internally oriented.

In addition to significant changes in indoor conditions, the organizational office layout also changed. Office space in the previous workplaces was organized in enclosed private offices, shared by several people. In the new building, the office space follows an open office layout. Open offices tend to generate noise complaints among occupants, who can be distracted by high levels of noise and loss of privacy. Indeed, a recent study shows workplace distraction significantly reduces worker productivity [21]. We therefore explore the changes in noise satisfaction in our analysis.

The newly constructed building is located very close to the existing buildings—two buildings are within 150 meters (a tenth of a mile), and equally close to public transit and highways ramps. The third building is located at a distance of 1km (0.6 mile), and next to a local train station, the main commuting mode for employees that use public transit (the new building is close to an intercity train station). Overall, employees that moved to the new building did not experience meaningful differences in commuting patterns, and if anything, conditions for access to public transit get slightly worse.

## 2.2 Survey design

We received permission from the municipality to send surveys to all of its employees, asking them to complete the survey via email (See S1 Appendix for the text of the invitation sent to the employees). The survey included anonymized individual identifiers, allowing us to build a longitudinal dataset that tracks the responses from the same employee over multiple survey waves. Our sample includes the survey responses of the treatment group, individuals relocated to the new "green" building, and those of the control group, comprising those employees who were not relocated to the new building.

The surveys span the period before and after the relocation. We first sent a questionnaire to all employees one month before the relocation took place, serving as the baseline survey in the analysis. After the relocation, we surveyed all employees three times, six months to two years after the relocation. We surveyed those individuals that were relocated as well as those who remained in their original workplace during the entire sample period. The three surveys cover both the cold and warm seasons (see Fig 1 for the exact timeline of the surveys).

The survey included the module developed by the Center for the Built Environment (CBE) at the University of California, Berkeley, to monitor the perceived environmental conditions of occupants in their workplaces [22]. Since the early 2000s, the CBE survey has been extensively used to evaluate the performance of buildings globally. The core questions in the survey assess occupant (dis)satisfaction and comfort with indoor environmental quality (IEQ) issues, including indoor air quality, thermal comfort, lighting, and acoustics. We asked participants to rate their satisfaction with different aspects of the indoor environment on a 7-point scale, ranging from "very satisfied" to "very dissatisfied," with a neutral midpoint. In a second set of questions, we asked participants to rate each IEQ dimension on 7-point scales ranging from "support" to "interfere" with their ability to get work done (see Panel A of Table A.1 in S1 Appendix).

The survey included two questions about the health status of individuals. First, we examined changes in the health status based on the prevalence of symptoms related to the Sick Building Syndrome (SBS). This concept is widely examined in the building science and public health literature and refers to "a collection of non-specific symptoms including eye, nose and throat irritation, mental fatigue, headaches, nausea, dizziness and skin irritations, which seem to be linked with occupancy of certain workplaces" [23]. The survey included a question asking whether the subject suffered from SBS symptoms ("Do you regularly have symptoms (e.g. tiredness, headache, eye irritation, nasal congestion, dry throat, dry skin) that disappear when you leave the building?"). In addition, we collected self-reported sick leave data based on the number of days missed due to health reasons in the year before the survey ("How many days were you unable to work this year due to illness?") in a categorical question where participants were asked to choose between the following options to report the number of sick days: (1) I did not report sick this year, (2) 1 day, (2) 2-5 days, (3) 5-10 days, and (4) more than 10 days.

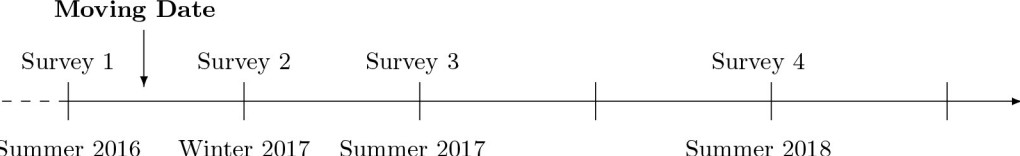

**Fig 1. Timing of survey waves.** The figure illustrates the exact timing of each survey wave. The first survey (k = 0) took place before the moving date (August 2016). The second (k = 1), third (k = 2) and fourth (k = 3) took place after the relocated group moved to the new building, in respectively January 2017, July 2017 and October 2018.

We measured the job satisfaction of employees based on a series of Likert scales, where respondents rated the frequency with which they experienced job-related emotions (see Table A.2 in S1 Appendix for the complete list of questions). The frequency of the scales ranged from "Daily" to "Never," and includes the options "A few times a week," "Once a week," "Few times a month," "Once a month," and "Few times a year or less." To help measure job satisfaction, we construct an index equal to the first principal component of all survey questions listed in the Table A.2 in S1 Appendix, together with the loadings of the first principal component of job satisfaction.

In addition, the survey included a series of questions that asked participants to grade their (dis)satisfaction with the layout, and furniture and equipment in their workplace, based on the same scales that were used to grade the perceived environmental conditions in the workplace (see Panel B of Table A.2 in S1 Appendix). We created dummies from the original scales that take the value of 1 if the original scale is "Daily," "A few times a week," or "Once a week," and zero otherwise. Finally, the survey included questions about basic demographic characteristics of respondents (i.e., age and gender) and some details of the employee's employment contract (i.e., working hours, and years at the current organization).

## 2.3 Descriptive statistics

The response rate of the surveys ranged between 35% and 40%. In the first wave, we gathered 573 valid answers, 585 in the second wave, 569 in the third wave, and 530 in the fourth wave. The median completion time of the survey was 11 minutes. We observe no differences in response time between relocated (treated) and non-relocated (control) employees, suggesting no differences in attention or effort between the two groups (but, of course, our goal is not to empirically assess such differences).

Table 1 shows the demographic characteristics of the relocated and non-relocated workers in wave 1 (i.e. the survey administered before the move). The non-relocated employees are younger, on average, than those in the relocated group, as reflected by the higher percentage of individuals below 31 years old (19% vs. 10%). The gender ratio does not differ between two employee groups.

Looking at the current contract characteristics of the two groups of employees, we find that the non-relocated individuals are less experienced than those in the relocated group (as reflected in the age difference between the groups). The total working hours do not differ significantly between the treatment and control group. Overall, the pre-trends reported in Table 1 indicate the absence of meaningful differences in the main outcomes of interest—the reporting of SBS symptoms, job satisfaction, and the incidence of sick leave.

Fig 2 provides simple non-parametric comparisons of the metrics that relate to occupant (dis)satisfaction and issues with indoor environmental quality (IEQ). Each chart shows the percentage of dissatisfied respondents on 10 dimensions of IEQ, over the four survey waves. A lower percentage implies a higher satisfaction with that particular IEQ dimension.

Panel A shows the satisfaction metrics for the non-treated sample, i.e. those survey respondents that remained in the same building. The scores on almost all IEQ dimension are exactly equal before and after the moving date, which is in line with expectations—after all, nothing changed for these survey respondents. There are some small improvements in satisfaction with two air quality dimension, but those improvements are not consistent across survey waves, perhaps indicating a seasonal effect.

Panel B shows the IEQ satisfaction for the treated sample. Quite clearly, the satisfaction of relocated employees with IEQ dimensions related to indoor air quality, temperature, and to a

**Table 1. Descriptive statistics sample in first survey wave (Before the move, July 2016).**

| | Non-Relocated (N = 247) | Relocated (N = 326) | Diff. |
|---|---|---|---|
| *Age* | | | |
| Below 31 Years Old (1 = Yes) | 0.19 | 0.10 | 0.09** |
| 31-50 Year Old (1 = Yes) | 0.34 | 0.45 | -0.11** |
| 50 Years Old or Older (1 = Yes) | 0.47 | 0.45 | 0.02 |
| *Gender* | | | |
| Female (1 = Yes) | 0.46 | 0.50 | -0.04 |
| *Health & Productivity* | | | |
| Sick Building Syndrome (1 = Yes) | 0.44 | 0.42 | 0.03 |
| No Days on Sick Leave (1 = Yes) | 0.53 | 0.53 | -0.01 |
| *Time Working for The Company* | | | |
| Less than 1 Year | 0.23 | 0.12 | 0.11*** |
| 1-2 Years | 0.38 | 0.24 | 0.14*** |
| 3-5 Years | 0.16 | 0.27 | -0.11** |
| More than 5 Years | 0.23 | 0.37 | -0.14*** |
| *Working Hours per Week* | | | |
| Less than 10 Hours | 0.06 | 0.03 | 0.03 |
| 11-30 Hours | 0.41 | 0.49 | -0.08 |
| More than 30 Hours | 0.53 | 0.48 | 0.05 |

Stars indicate the significance of the p-values of t-tests of differences in means across groups.

p<0.05,

** p<0.01,

*** p<0.001

lesser extent views and light, increased after the move took place. Concomitantly, employee satisfaction with sound and privacy-related IEQ dimensions decreased slightly.

## 3 Methods

### 3.1 Empirical strategy

We use difference-in-difference (DiD) models to estimate the impact of the improvement in building conditions on employees' perceived working conditions and health status. The DiD research design relies on the assumption that the characteristics of workers who were relocated to the new building changed over time in a way that is comparable to those who were not relocated. To alleviate concerns of potential biases in our results, we estimate our parameter of interest in a regression model with a rich set of fixed effects and time-varying control variables:

$$Y_{it} = \mu_i + \tau_t + \delta Relocated * AfterMove_{it} + \beta X_{it} + \epsilon_{it} \tag{1}$$

where $Y_{it}$ includes the set of outcome variables describing the perceived working conditions and health status of individual $i$ at time $t$. We include the scales describing the perceived noise, temperature, light, and air quality in the workplace. Finally, we consider a dummy variable indicating whether the individual suffers from SBS.

Our prime parameter of interest is $\delta$, describing the average change in the outcomes ($Y_{it}$) after the move for the employees who relocated to the new building. The individual fixed effects ($\mu_i$) should reduce bias resulting from differences between the movers and non-movers.

### a. Individuals in Original Buildings

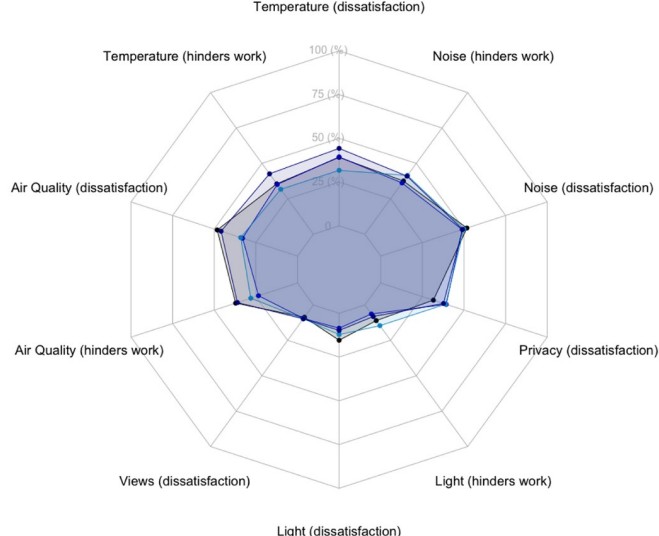

### b. Individuals Relocated to New Building

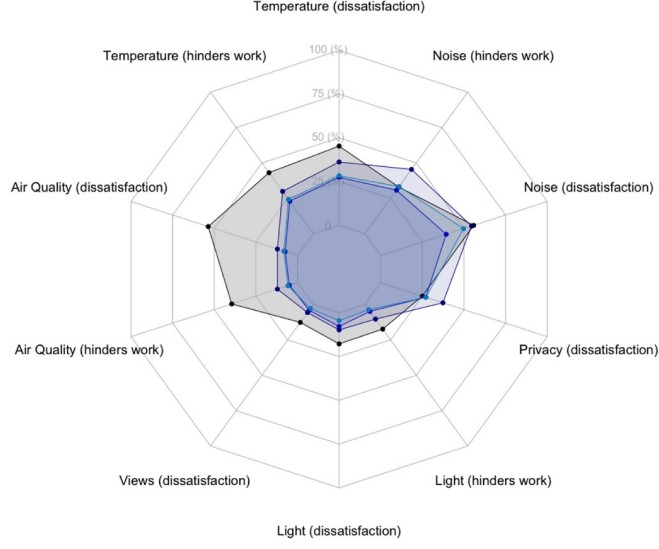

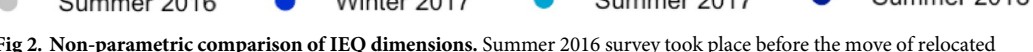

**Fig 2. Non-parametric comparison of IEQ dimensions.** Summer 2016 survey took place before the move of relocated workers to the new building. Percentage based on the number of respondents that rate the environmental dimensions as larger than four on a seven-point Likert scale—ranging from satisfied (1) to dissatisfied (7).

In addition, we include time dummy variables $\tau_t$ for each survey wave, non-parametrically adjusting for possible shocks in the city or employer that coincide with the move (e.g., pollution reduction in the city). Finally, we include a set of individual time-varying controls, $X_{it}$. The set of controls includes the average working hours per week and the reported scales rating the *Office Layout* (See Panel A of Table A.1 in S1 Appendix for the full list of scales in this category). $\epsilon_{it}$ is the error term, which might be correlated within individuals. Therefore, we cluster standard errors at the individual level.

We then use an event-study analysis to capture dynamic effects of the new building on the employees. Eq 2 estimates the effects of the relocation separately by year:

$$Y_{it} = \mu_i + \tau_t + \sum_{k=1}^{K} \delta_k Relocated * AfterMove_{it}^k + \beta X_{it} + \epsilon_{it} \tag{2}$$

Here, the coefficient $\delta_k$ describes the effect of working in the newly constructed office $k$ periods after the moving date. Thus, $Relocated * AfterMove_{it}^k$ is an indicator for being $k$ time periods relative to the moving date. The reference category is $k = 0$; hence, the post-treatment effects are relative to the year immediately before the treated individuals were relocated to the new building.

In a final step, we estimate to what extent the changes in each of the environmental scales with respect to their baseline level translate into changes in health status with respect to the baseline:

$$Health_{it} - Health_{ib} = \tau_t + \Theta_s(IEQ_{its} - IEQ_{ibs}) + \beta(X_{it} - X_{ib}) + u_{it} - u_{ib} \tag{3}$$

where $Health_{it}$ takes the value of 1 if individual $i$ reports SBS symptoms at time $t$, and zero otherwise. $Health_{ib}$ takes the value of 1 if individual $i$ reports SBS symptoms in the baseline survey. $Health_{it} - Health_{ib}$ describes the difference between individual $i$'s probability of stating SBS status at time $t$ and in the baseline survey $b$. Similarly, $IEQ_{its} - IEQ_{ibs}$ describes the changes in the values reported in the environmental scale $s$ for individual $i$ at time $t$, with respect to his answers in the baseline survey $b$. The coefficients of interest, $\Theta_s$, describe how changes in environmental scale $s$ translate into changes in the probability of reporting SBS symptoms. In addition, we include the changes in a set of control variables for building quality $X_{it} - X_{ib}$. Error terms are clustered again at the individual level.

## 3.2 Ethics statement

The Gemeentesecretaris of Venlo vetted the concept survey designed by Maastricht University in the Summer of 2016. Under Dutch law, this is the highest ranking civil servant of a city, with overall responsibility for all matters pertaining to the city bureaucracy, including privacy issues of personnel. He gave his consent, conditional upon agreement by the city's privacy controllers and his fellow directors. Consequently, the survey, recruitment scripts, and informed consent protocols were sent to the privacy controllers of the city of Venlo for further revisions and vetting. The privacy controllers gave their consent, and the survey went for a final round of vetting to the Board of Directors of the city bureaucracy. The survey was distributed after the directors gave final permission. The distribution of the survey, and all the communication with the subjects in the study was made directly by members of the local government, via a corporate email account. The survey responses were directly loaded in an online platform, where researchers could access the data for further analysis. No human subject was directly contacted by the research team. The research team had no access to information through which survey participants could be identified.

## 4 Results

### 4.1 Difference-in-differences

In this section, we report the estimated coefficients and standard errors of the coefficients associated with the DiD variable in Eq 1. Table 2 provides the estimation results. Column (1) shows the estimated DiD coefficients, including time dummies, individual-fixed effects, and time-varying controls. We include changes in working hours, perceived quality of furniture, and office layout as time-varying controls.

The estimation results indicate that the relocation of employees is associated with a significant decrease in the level of dissatisfaction of perceived environmental quality in all measures, except for noise and privacy. The highest impact associated with the relocation is on the air quality dimension, where the dissatisfaction scale drops 1.4 points on a 7-point scale. In

**Table 2. Estimation results from difference-in-difference model.**

| | | (1)<br>Full<br>Sample | (2)<br>Men | (3)<br>Women | (4)<br>Age<br>Below 31 | (5)<br>Age<br>30-50 | (6)<br>Age<br>Above 50 |
|---|---|---|---|---|---|---|---|
| *Indoor Environmental Conditions* | | | | | | | |
| Air Quality | Dissatisfaction | -1.443*** | -1.622*** | -1.545*** | -1.766** | -1.439*** | -1.677*** |
| | | (0.163) | (0.256) | (0.229) | (0.627) | (0.280) | (0.231) |
| | Hinders Work | -1.091*** | -1.077*** | -1.322*** | -1.615*** | -1.516*** | -1.125*** |
| | | (0.149) | (0.232) | (0.198) | (0.432) | (0.267) | (0.215) |
| Temperature Quality | Dissatisfaction | -0.593** | -1.070*** | -0.437 | -0.794 | -0.707* | -0.698** |
| | | (0.195) | (0.263) | (0.272) | (0.737) | (0.302) | (0.267) |
| | Hinders Work | -0.529** | -1.056*** | -0.359 | -1.041 | -0.624 | -0.741** |
| | | (0.182) | (0.260) | (0.233) | (0.608) | (0.339) | (0.231) |
| Light Quality | Dissatisfaction | -0.503** | -0.607** | -0.720** | -0.871 | -0.038 | -0.871*** |
| | | (0.157) | (0.191) | (0.223) | (0.530) | (0.228) | (0.184) |
| | Hinders Work | -0.435** | -0.473* | -0.695*** | -1.531** | -0.172 | -0.677*** |
| | | (0.150) | (0.190) | (0.207) | (0.480) | (0.208) | (0.192) |
| Views | Dissatisfaction | -0.338* | -0.560** | -0.544** | -1.454* | -0.233 | -0.478* |
| | | (0.146) | (0.212) | (0.196) | (0.601) | (0.180) | (0.192) |
| Noise | Dissatisfaction | 0.211 | 0.014 | 0.097 | 1.006* | -0.104 | 0.006 |
| | | (0.158) | (0.216) | (0.196) | (0.486) | (0.229) | (0.206) |
| | Hinders Work | 0.069 | -0.155 | 0.065 | 1.130** | -0.127 | -0.208 |
| | | (0.148) | (0.199) | (0.191) | (0.396) | (0.219) | (0.199) |
| Privacy | Dissatisfaction | -0.024 | -0.318 | -0.014 | 0.470 | -0.158 | -0.397 |
| | | (0.171) | (0.230) | (0.245) | (0.645) | (0.263) | (0.239) |
| *Health Indicator* | | | | | | | |
| Sick Building Syndrome | Dummy (1 = Yes) | -0.216*** | -0.277*** | -0.223** | -0.355 | -0.216* | -0.255** |
| | | (0.056) | (0.075) | (0.078) | (0.223) | (0.085) | (0.078) |
| Wave-Fixed Effects | | YES | YES | YES | YES | YES | YES |
| Individual-Fixed Effects | | YES | YES | YES | YES | YES | YES |
| Controls | | YES | YES | YES | YES | YES | YES |

Robust standard error clustered at the individual level.

* $p<0.1$,

** $p<0.05$,

*** $p<0.01$.

Controls in Column (5) include the satisfaction scales for layout and furniture scales listed in Table in S1 Appendix and average working hours per week.

relative terms, when compared to the average value of these scales in the baseline survey for the relocated group, the relocation to the new building improve employee satisfaction with air quality by 32% (1.44/4.50). Similarly, the relocated employees attach a 26% (1.09/4.14) lower value to the scale evaluating whether air quality hinders work.

The relocation of workers to the sustainable building also generates significant improvements in the perception of light quality and temperature. The absolute and relative improvement in the scales of these two parameters is smaller than the changes observed for the air quality dimension. Temperature dissatisfaction among relocated workers drops, on average, by 0.59%, that is 17% compared to the average value of the relocated group before the move. Similarly, the relocation reduces dissatisfaction with the light quality in the building by 0.5%, that is 28% compared to the average value of the relocated group before the move.

Anecdotally, we can confirm the enhanced quality of the indoor environmental conditions in the new building through the use of indoor air quality data from several sensing campaigns organized by the municipality in all of their buildings. While a formal analysis is not possible, given the limited scope of the sensing data, Fig A.1 in S1 Appendix provides a summary of $CO_2$ levels and maximum temperature in one of the existing buildings (2016) versus the new building (2017), during periods of comparable outdoor conditions. The $CO_2$ measurements, commonly used in the building science literature to evaluate the performance of ventilation systems, corroborate that the air quality of the "green" building relative to the old building is significantly better. Furthermore, the temperature data indicate that, during hot days (peak day above 30°C), there are substantial improvements in the thermal conditions in the green building, which manages to keep the indoor temperature in an optimal range of 22-24°C (vs. 26-29°C in the old building).

Importantly, we observe significant improvement in the health of individuals. Column (1) of Table 2 shows a decrease of 21.6% in the prevalence of symptoms related to the SBS among the relocated workers (after the move). The relocation of workers generates a substantial drop in the prevalence of SBS symptoms when compared to the baseline probability of reporting SBS symptoms among the relocated employees, by 42%.

## 4.2 Dynamic effects

We test for the existence of a possible rebound in the improvement of perceived health and well-being experienced by the relocated employees. Evidence from psychology and behavioral economics shows that individuals tend to adapt, in the medium term, to changes in their physical or material conditions [e.g., 24]. Thus, the estimated changes in the subjective assessments presented in the previous section might be biased by an overreaction of individuals in the short term. In addition, the potential material depreciation of the new building might also distort the results.

Fig 3 shows the coefficients ($\hat{\delta}_k$) describing the changes in responses across survey waves ($k$) with respect to the baseline survey for the relocated (blue) and non-relocated employees (gray), together with their associated confidence intervals. Our surveys are taken from six months to two years after the relocation. In addition, the three surveys cover both the cold and warm seasons. Overall, we observe stability in the coefficients describing the changes in health status over time. We find no evidence of a rebound in the estimated changes in the health status of the employees. The estimation results indicate the initial drop in the prevalence of SBS symptoms remains at that level ($\delta_{k=1} = 0.21$).

## 4.3 Heterogeneous effects

In this section, we study whether some subgroups are more sensitive to indoor environmental conditions than others. First, we explore gender differences. Current regulations regarding

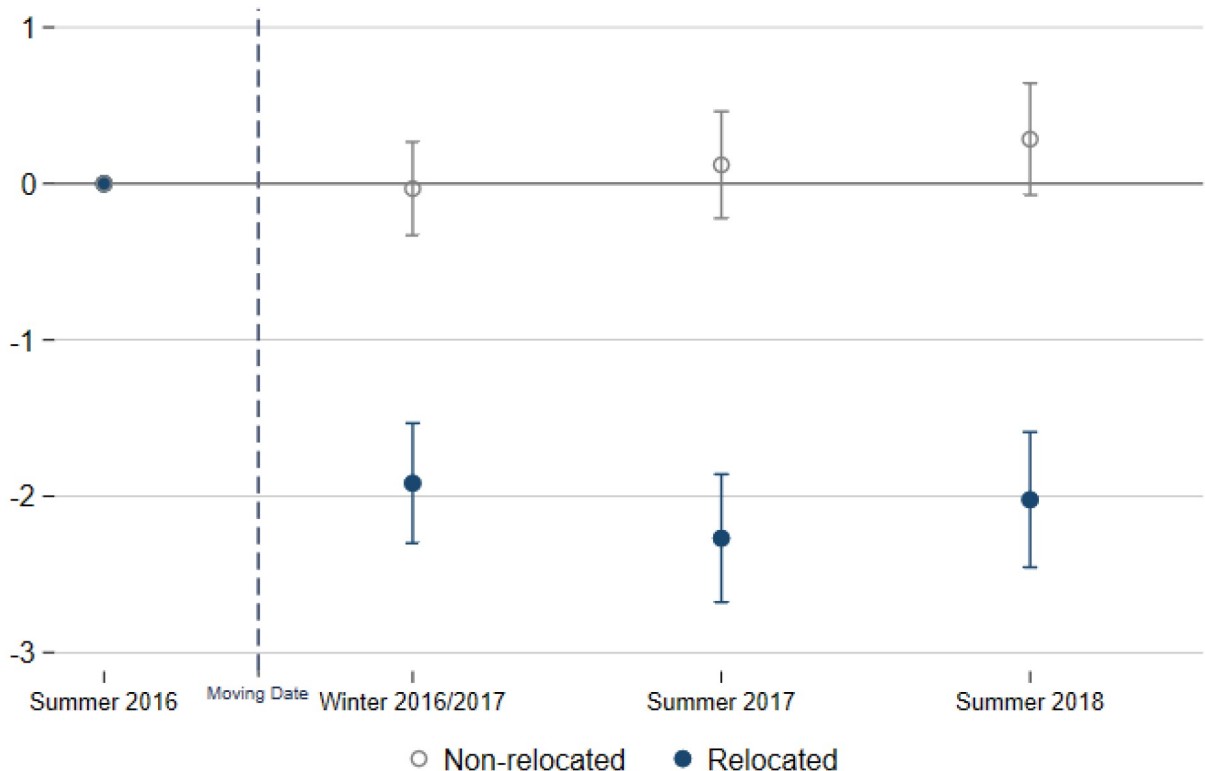

**Fig 3. Trends in sick building syndrome.** The figure shows the estimated coefficient of the time dummies denoting the survey waves before (= 0) and after the moving date. The dots represent the point estimates and the bars the 95% confidence intervals. The vertical, dashed gray line indicates the moving date. The set of control variables includes the average hours worked per week and the layout scales.

indoor climate conditions in office buildings tend to be based on a thermal comfort model developed in the 1960s. That model optimizes the environmental conditions to satisfy an average male. A recent study in biophysics indicates the existing model significantly miscalculates the metabolic rate of female thermal demand [25]. This is line with many field studies showing that females express more dissatisfaction than males with low temperatures [for a review of the literature, see 26]. In addition, the presence of pre-existing diseases in the respiratory or cardiovascular systems among older population groups might exacerbate the health impacts of certain hazards in the indoor environment (e.g., indoor pollutants) [27]. For the analysis of the differences across demographic groups, we therefore stratify our sample by gender and age.

Table 2, Columns 2 to 6 presents the results of the heterogeneity analysis. Columns 2 and 3 display the results for the two gender subsamples, and columns 4 to 6 show the estimates for the three age groups in our sample. We observe no significant discrepancies in responses to scales in the noise, air, and light quality dimensions across gender or age groups. However, the results for thermal dissatisfaction indicate the drops in dissatisfaction rates associated with the new building are present only among male employees. Relocated women did not significantly adjust their ratings after being transferred to the new building. Similarly, we observe significant changes in thermal dissatisfaction among the older employees only (beyond 30 years old).

When focusing on the health measures, we find the relocation of workers to the new building generates similar drops in the prevalence of SBS symptoms for female and male employees. The estimates suggest the impact of the relocation becomes more significant with the age of employees. We observe significant drops in the probability of reporting SBS among the oldest

group of employees only (over 50 years old). The coefficient associated with the group of workers between 30 and 50 years old is marginally significant (i.e., at the 10% level).

## 4.4 Health and productivity

The relocation to a new building involves significant changes in a variety of factors regarding the workplace of employees. We therefore analyze how the changes in different dimensions of indoor environmental conditions contributed to the drop in the prevalence of SBS symptoms, with respect to the initial situation just before the moving date.

Fig 4 presents the estimated coefficients $\Theta_s$ of Eq 3, describing the association between the probability of reporting SBS symptoms and each of the four factors related to the indoor environmental conditions in the workplace (i.e., elements of $\Theta$). The estimation results indicate that poor air quality is, on average, the only significant contributor to the prevalence of SBS symptoms. The presence of perceived deficient air quality is associated with an increase of about 10 percentage points in the odds of reporting SBS in our sample.

We also include a series of furniture quality factors as regressors in the empirical model, as placebos to construct a falsification test. The absence of significant coefficients associated with these factors supports the hypothesis that the health improvements displayed in this study are mainly driven by an improvement in environmental conditions in the workplace and not by a general building quality improvement.

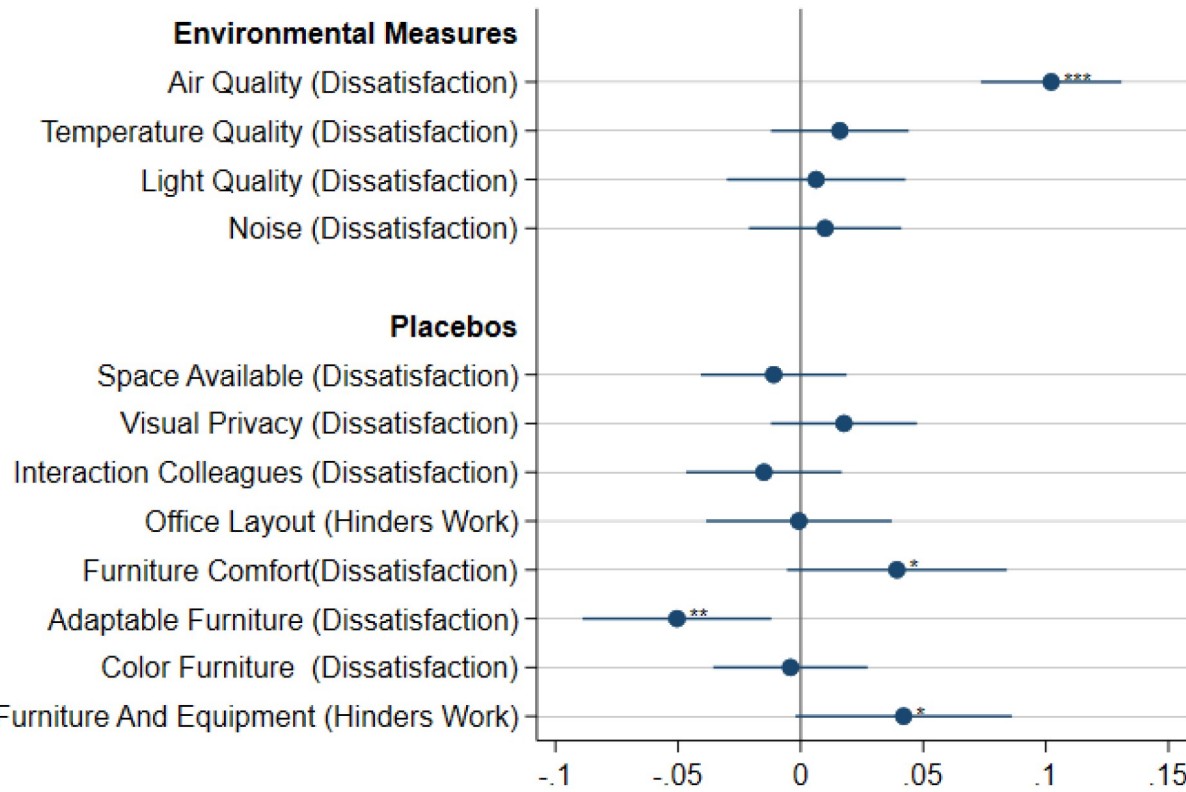

**Fig 4. Effect explaining sick building syndrome by IEQ dimension.** The figure shows the point estimates of and confidence intervals associated with each ot the elements of vector $\Theta$ in Eq 3. The dots represent the point estimates and the bars the 95% confidence intervals. All regressions include time varying controls (contract type), individual and survey-wave fixed effects. * p<0.1, ** p<0.05, *** p<0.01.

## a. Relation and Job Satisfaction

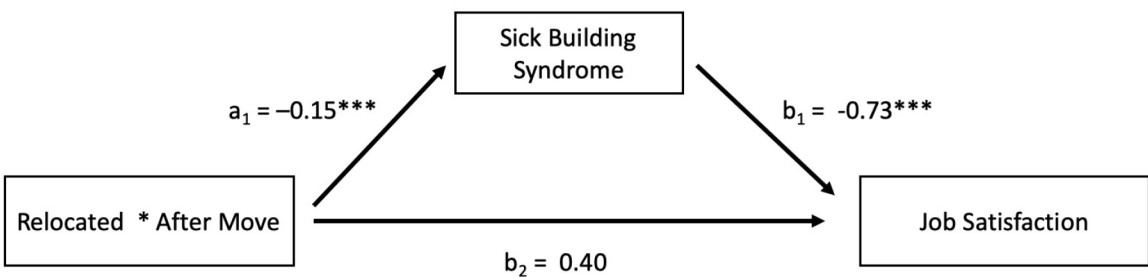

## b. Relocation and Sick Leave

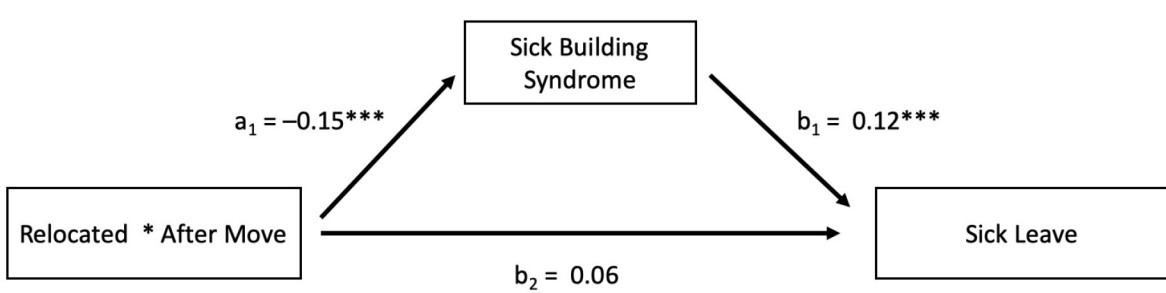

**Fig 5. Impact of relocation on productivity.** The figure shows sick building syndrome (SBS) as a mediator of the effect of the relocation to the new healthy building. The regressions include a dummy variable indicating whether the individual was in the relocated group, and survey-wave fixed effects. $^*$ p<0.1, $^{**}$ p<0.05, $^{***}$ p<0.01. The mediation equations are: $Y_{it} = \tau_1 \, After\, Move_t + \mu_1 \, Relocated_i + b_1 \, SBS_{it} + b_2 \, Relocated \, ^* \, After\, Move_{it} + \epsilon_{it1}$ (4) $SBS_{it} = \tau_2 \, After\, Move_t + \mu_2 \, Relocated_i + a_1 \, Relocated \, ^* \, After\, Move_{it} + \epsilon_{it2}$ (5) Where $Y_{it}$ denotes the sick building syndrome (SBS) or job satisfaction of individual i in survey wave $t$.

We then implement a mediation analysis [15; 28] to test for the impact of the relocation on job satisfaction and employee sick leave, where the SBS symptoms act as mediator. For the analysis, we construct a dummy variable that takes the value of one if the individual reports sick, and zero otherwise—focusing on the extensive margin. For job satisfaction, we construct an index equal to the first principal component of all job satisfaction survey questions listed in Table A.2 in S1 Appendix. The index ranges from -3.20 to 14.75.

Fig 5 Panel A shows that effect of the relocation on job satisfaction is mediated by the presence of SBS symptoms. The first, direct path indicates that there is no evidence of a direct effect of the relocation on job satisfaction of employees. However, the estimates from the second, indirect path indicate the presence of a significant average causal mediation effect of 0.11, with a 95% confidence interval that does not include zero [.20, .03]. The increase in job satisfaction corresponds to 4% of a standard deviation in the constructed index.

Fig 5 Panel B shows that effect of the relocation on sick leave is also mediated by the presence of SBS symptoms. The first, direct path again indicates that there is no evidence of a direct effect of the relocation on the incidence of sick leave of employees. However, the estimates from the second, indirect path indicate the presence of a significant average causal mediation effect of -0.02, with a 95% confidence interval that does not include zero [-.04, -.01]. This result implies that the mediating hypothesis is supported. The relocation of individuals to the healthy building leads to a drop in the prevalence of sick leave of 2%, with the presence of symptoms related to SBS as the main mechanism for explaining the treatment effects.

## 5 Conclusions

Human health is a critical factor for the generation of output by workers. Companies bear substantial costs from the absenteeism and presenteeism among their employees [1; 2]. In addition, increasing evidence shows job satisfaction translates into higher productivity for workers and ultimately higher value for companies [29].

This study investigates the impact of the indoor environmental conditions in the workplace on the health and job satisfaction of employees, as core factors of productivity. We exploit a natural experiment, based on the relocation of 70% of the workforce of a municipality in the south of the Netherlands. The estimation results show a significant improvement in the perceived indoor environmental conditions and health of the relocated workers. We find the largest improvements in perceived air quality of the workplace, reducing the level of dissatisfaction by 1.62 points on a 7-point Likert scale. In addition, we observe significant improvements in the health status of individuals. In particular, we observe a 42% reduction in the prevalence of SBS symptoms.

The results of the heterogeneity analysis show the existence of differences among workers. The relocation to the new building had significant effects on the perceived environmental conditions of men and not women. However, the relocation led to a drop in SBS symptoms that was similar in magnitude between men and women. The analysis of different age subsamples indicates increasing benefits of good environmental quality with age. Older individuals benefited the most from the move in terms of perceived environmental quality and health status.

Importantly, a mediation analysis shows that the relocation had a significant impact on two proxies for productivity. Job satisfaction increased, and there was a drop in the prevalence of sick leave by 2%. While we don't have access to administrative records, an overly simplified back-of-the-envelop calculation shows that this reduction has meaningful implications for the public finances of the municipality. In 2019, the aggregate salary spent on employees working in the new building was €54 million, for a total of 800 employees. Of those 800 employees, 43% report in sick at least once per year, with an average sick leave of about 5 days per employee per year. The reduction of sick leave by 2% (considering solely the extensive margin, not the changes in length of sick leave), saves the municipality €25,000 per year, which seems small, but at current cost of capital (around 1%) translates into a capitalized value of €2.5 million.

Of course, the improvement in productivity through reduced sick leave is just a "co benefit" of the newly constructed building. The city of Venlo did an extensive cost-benefit analysis for the new "green" building. This life-cycle analysis calculated both costs and benefits for a 40-year period. The outcomes of this analysis show a €3.4 million marginal investment in technical installations and healthy materials, in addition to the budget that would have been needed for a conventionally engineered building. The design created savings in building management and exploitation costs, for example maintenance and energy, of approximately €17 million over the 40-year period. Importantly, this cost-benefit analysis only involved proven benefits. Given the uncertainty of any productivity and/or health benefits of the new building (convincing academic evidence for this was lacking at the time), these were not included in the analysis.

The results in this paper add to a growing body of research on the implications of buildings on health and productivity outcomes, where most studies are based on engineering measures or on cross-sectional analyses. The quality of indoor environmental conditions may have significant economic implications for our service society, which depends on buildings in order for workers to be productive. Our findings show that variation in different dimensions of

indoor environmental quality affect perceived health outcomes, which has implications not just on worker productivity, but also on the cost of absenteeism. Of course, the reported changes in environmental conditions and health and well-being are based on *perceptions* of employees. While the objective measure of sick leave clearly shows the effect of enhanced of indoor environmental quality, more research is needed on objective measures of employee productivity, including quantitative data on physical and mental health and well-being.

## Supporting information

**S1 Appendix.**
(PDF)

**S1 File.**
(ZIP)

**S2 File.**
(ZIP)

## Acknowledgments

We thank the editor and an anonymous referee for helpful comments. Michel Weijers (municipality of Venlo) provided excellent support in data collection and enabling this research. Authors are responsible for all errors.

## Author Contributions

**Conceptualization:** Juan Palacios, Piet Eichholtz, Nils Kok.

**Data curation:** Juan Palacios.

**Formal analysis:** Juan Palacios.

**Funding acquisition:** Nils Kok.

**Methodology:** Juan Palacios.

**Writing – original draft:** Juan Palacios, Piet Eichholtz.

**Writing – review & editing:** Nils Kok.

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
