## [Decision Letter · Decision Letter 0]

22 Apr 2020

PONE-D-20-00869

Moving to Productivity: The Benefits of Healthy Buildings

PLOS ONE

Dear Dr. Kok,

Thank you for submitting your manuscript to PLOS ONE. After careful consideration, we feel that it has merit but does not fully meet PLOS ONE’s publication criteria as it currently stands. Therefore, we invite you to submit a revised version of the manuscript that addresses the points raised during the review process.

The two referees have split suggestions. The second referee is a labor economist and raised a few valid concerns. Please try your best to address those concerns as much as you can.

I enjoyed reading your paper but also have a few comments for your reference.

You used the word “quasi-experiment”. I understand people often use natural experiment and quasi experiment interchangeably; I would suggest you use natural experiment. The reason is that relocating to the new building is a decision exogenous to individuals; it is probably a decision by the city government (please clarify if this is true). So for individual workers, relocation is an exogenous shock out of their control, and also the relocation itself is not for the purpose of making workers more healthy, it is probably due to the office space constraint or government’s budget situation. A quasi experiment means the setting is an experiment designed for a particular research but for some reasons the design cannot randomly assign subjects to the treatment group. A good reference is chapter 15 “Natural and quasi experiments” in a book “Research methods in practice” 2^nd^ edition by Remler and Van Ryzin, Sage Publications, 2015.The key of your research setting is to explain clearly how the relocation decision is made.  You only vaguely mentioned the selection is based on teams not individuals. More details are needed as the second referee is concerned about. For example, if a team is selected, are all members of the team relocated? Are there substantial differences between selected teams and those that did not move? These details do not jeopardize your DID identification strategy as your setting is a natural experiment. This also means you need to describe the background of the new office building and relocation (related to my previous comment) to show it is a natural experiment.What is the distance between the old and new office building? Does this affect workers’ commute behavior?Describe figure 1 the survey waves in section 2.2.Typo: first sentence of the 2^nd^ paragraph on p. 9: Figure 3 should be “Figure 2”.**A recent study shows workplace distraction significantly reduces worker productivity. You may cite it. (Citation:** Bialowolski P, McNeely E, VanderWeele TJ, Weziak-Bialowolska D (2020) Ill health and distraction at work: Costs and drivers for productivity loss. PLoS ONE 15(3): e0230562. https://doi.org/10.1371/journal.pone.0230562)

We would appreciate receiving your revised manuscript by Jun 06 2020 11:59PM. To enhance the reproducibility of your results, we recommend that if applicable you deposit your laboratory protocols in protocols.io, where a protocol can be assigned its own identifier (DOI) such that it can be cited independently in the future. For instructions see: http://journals.plos.org/plosone/s/submission-guidelines#loc-laboratory-protocols

We look forward to receiving your revised manuscript.

Kind regards,

Shihe Fu, Ph.D.

Academic Editor

PLOS ONE

Journal Requirements:

3. Your ethics statement must appear in the Methods section of your manuscript. If your ethics statement is written in any section besides the Methods, please move it to the Methods section and delete it from any other section. Please also ensure that your ethics statement is included in your manuscript, as the ethics section of your online submission will not be published alongside your manuscript.

Reviewers' comments:

Reviewer's Responses to Questions

**Comments to the Author**

1. Is the manuscript technically sound, and do the data support the conclusions?

Reviewer #1: Yes

Reviewer #2: Partly

2. Has the statistical analysis been performed appropriately and rigorously? 

Reviewer #1: I Don't Know

Reviewer #2: Yes

3. Have the authors made all data underlying the findings in their manuscript fully available?

Reviewer #1: Yes

Reviewer #2: Yes

4. Is the manuscript presented in an intelligible fashion and written in standard English?

Reviewer #1: Yes

Reviewer #2: Yes

5. Review Comments to the Author

Reviewer #1: This is a well-written analysis, definitely an important case study. It adds significantly to the sparse studies of the potential of green/sustainable building practices to affect user outcomes. More studies like this need to be done, so designers and clients can make better decisions.

Reviewer #2: This paper use difference in difference method to study how a reallocation of part of a municipality’s workforce to a green building affect the workers’ reported health status and job satisfaction. For this paper, I have the following questions and concerns.

(1) The health status used in this article are subjective measures, reported by employees. People’s subjective observation could be biased due to the building is new and everything looks good, instead of due to improvement in specific dimensions of the environment. Take the temperature quality as an example. The level and volatility of temperature is objectively measurable. The distinction between subjective and objective improvement in the environment is important for policy implications: is it good enough to make the environment “look green” or should the investment be spent on improving the air, temperature, and etc.

(2) How “green” is the new building? Is there a LEEDS rating for it? Is the effects showed here idiosyncratic to the building studied in this paper or can we extrapolate the effects to other buildings “similarly green”?

(3) The authors motivate the study with the tradeoff between the costs associated with improving working environment (say making office building green) and the costs associated with absenteeism (potentially due to sick-building-syndrom). Yet, in the analysis part, these costs and benefits are not explicitly discussed.

(4) Spillover effects of green building on the neighborhood, on CO2 emission and etc. is a bigger component of the benefits of green building and at the center of the decision about whether private parties should invest in making building green and whether the public/the government should subsidize making building green. The authors need to justify further why they focus on workers subjective feeling of health.

(5) “Job-related” motion are not only affected by physical environment but also by social environment. Is it possible that workers take reallocation as a chance to choose new “working neighbor” and thus improve their job-related emotions?

(6) Why not studying the mediation effects of subjective health on absenteeism? For example, regress absenteeism on both the treated*after and the subjective health, to see whether subjective health is the mechanism through which reallocation to a green building influence absenteeism.

(7) While the authors argue that the decision to move is random, the sample who respond to authors’ surveys may not be random. It is possible that people who have more emotions to express and or those who have more free time are more likely to respond to the authors survey. This may bias the estimation.

6. PLOS authors have the option to publish the peer review history of their article (what does this mean?). If published, this will include your full peer review and any attached files.

Reviewer #1: No

Reviewer #2: No

---

## [Decision Letter · Decision Letter 1]

29 Jun 2020

Moving to Productivity: The Benefits of Healthy Buildings

PONE-D-20-00869R1

Dear Dr. Kok,

We’re pleased to inform you that your manuscript has been judged scientifically suitable for publication and will be formally accepted for publication once it meets all outstanding technical requirements.

Kind regards,

Shihe Fu, Ph.D.

Academic Editor

PLOS ONE

Additional Editor Comments (optional):

Reviewers' comments:

Reviewer's Responses to Questions

**Comments to the Author**

1. If the authors have adequately addressed your comments raised in a previous round of review and you feel that this manuscript is now acceptable for publication, you may indicate that here to bypass the “Comments to the Author” section, enter your conflict of interest statement in the “Confidential to Editor” section, and submit your "Accept" recommendation.

Reviewer #1: All comments have been addressed

Reviewer #2: All comments have been addressed

2. Is the manuscript technically sound, and do the data support the conclusions?

Reviewer #1: Yes

Reviewer #2: Yes

3. Has the statistical analysis been performed appropriately and rigorously? 

Reviewer #1: I Don't Know

Reviewer #2: Yes

4. Have the authors made all data underlying the findings in their manuscript fully available?

Reviewer #1: Yes

Reviewer #2: Yes

5. Is the manuscript presented in an intelligible fashion and written in standard English?

Reviewer #1: Yes

Reviewer #2: (No Response)

6. Review Comments to the Author

Reviewer #1: (No Response)

Reviewer #2: The author has addressed my questions and comments properly. I recommend this article for publication.

7. PLOS authors have the option to publish the peer review history of their article (what does this mean?). If published, this will include your full peer review and any attached files.

Reviewer #1: No

Reviewer #2: No

---

## [Editor Report · Acceptance letter]

16 Jul 2020

PONE-D-20-00869R1 

Moving to Productivity: The Benefits of Healthy Buildings 

Dear Dr. Kok:

I'm pleased to inform you that your manuscript has been deemed suitable for publication in PLOS ONE. Congratulations! Your manuscript is now with our production department. 

Kind regards, 

on behalf of

Dr. Shihe Fu 

Academic Editor

PLOS ONE